# Impact of the COVID-19 Pandemic on Tuberculosis Control: An Overview

**DOI:** 10.3390/tropicalmed5030123

**Published:** 2020-07-24

**Authors:** Kefyalew Addis Alene, Kinley Wangdi, Archie C A Clements

**Affiliations:** 1Faculty of Health Sciences, Curtin University, Bentley, WA 6102, Australia; archie.clements@curtin.edu.au; 2Wesfarmers Centre of Vaccines and Infectious Diseases, Telethon Kids Institute, Perth, WA 6009, Australia; 3Department of Global Health, Research School of Population Health, Australian National University, Canberra, ACT 2601, Australia; kinley.wangdi@anu.edu.au

**Keywords:** COVID-19, pandemic, endemic, tuberculosis, impact, control, overview

## Abstract

Throughout history, pandemics of viral infections such as HIV, Ebola and Influenza have disrupted health care systems, including the prevention and control of endemic diseases. Such disruption has resulted in an increased burden of endemic diseases in post-pandemic periods. The current coronavirus disease 2019 (COVID-19) pandemic could cause severe dysfunction in the prevention and control of tuberculosis (TB), the infectious disease that causes more deaths than any other, particularly in low- and middle-income countries where the burden of TB is high. The economic and health crisis created by the COVID-19 pandemic as well as the public health measures currently taken to stop the spread of the virus may have an impact on household TB transmission, treatment and diagnostic services, and TB prevention and control programs. Here, we provide an overview of the potential impact of COVID-19 on TB programs and disease burden, as well as possible strategies that could help to mitigate the impact.

## 1. Historical Perspective

Tuberculosis (TB) is one of the oldest endemic diseases affecting humanity, but it remains a significant global public health problem today [1,2]. It is estimated that a quarter of the world’s population has latent TB infection (a dormant form of TB) [3]. According to the World Health Organization (WHO) report, an estimated 10 million people develop active TB and more than one million people die due to TB each year [4].

The burden of TB has varied across human history. Major disruptions such as natural disasters, war and infectious disease pandemics have compromised TB programs and led to an increased burden of TB. For instance, during the First and Second World Wars, there were epidemics of TB in many European countries, which accounted for nearly one fourth of the total deaths during this period [5,6].

After World War II, following the discovery of TB medications, and improvements in the socioeconomic status of the population, TB was well controlled in high-income settings with TB burden decreasing dramatically [7]. However, when human immunodeficiency virus (HIV) emerged as a worldwide pandemic disease in the 1980s, TB re-emerged as an opportunistic infection and killed millions of people [8]. With the implementation of several TB prevention programmes and the introduction of anti-retroviral therapy (ART) for HIV, the mortality and morbidity of TB have decreased gradually over the last few decades [9].

More recently, the emergence of major viral disease outbreaks in several parts of the worlds has posed new challenges for global and national TB control efforts. For instance, the recent outbreak of Ebola in West Africa has severely compromised TB programs in the affected countries [10,11]. The outbreaks of Middle East respiratory syndrome coronavirus (MERS-CoV) complicated the control of TB in Saudi Arabia [12]. The direct and indirect effects of these viral outbreaks on TB programs have resulted in an increased burden of TB in the affected regions in subsequent years.

The United Nations Sustainable Development Goals (SDGs) include ending the TB epidemic by 2030, and the WHO set ambitious targets, including a 90% reduction in TB incidence and a 95% reduction in TB deaths compared with 2015, and no catastrophic costs due to TB, by 2035 [13]. Although progress continues to be made to achieve these ambitious targets, the current pandemic of coronavirus disease 2019 (COVID-19) could be a major challenge to achieving them.

Given the high levels of global disruption caused by the COVID-19 pandemic, it is critical to consider the potential impact on the control and prevention of common endemic diseases that might be even more devastating to human health than COVID-19 itself. The impact of COVID-19 on other diseases such as cancer and diabetes has been addressed in recent reviews [14,15]. However, there are no published reviews on the impacts of COVID-19 on TB. TB is the leading cause of death due to an infectious disease globally, and it is anticipated that people ill with both TB and COVID-19 may have poorer treatment outcomes. Most importantly, the public health response to COVID-19 to isolate people in their homes for extended periods could facilitate transmission of TB since close household contact, particularly in low-socioeconomic and overcrowded conditions, is a key risk factor for TB. Therefore, understanding the potential impacts of COVID-19 on TB is important for designing prevention strategies. Therefore, we review the potential impact of COVID-19 pandemics on the prevention and control of TB and provide possible strategies that could help to mitigate the impact.

## 2. The Coronavirus Disease 2019 (COVID-19) Pandemic and TB

The current pandemic of COVID-19 is a global health crisis, causing substantial disruptions to healthcare systems, including TB programs [16]. This section briefly summarizes the clinical features, epidemiological distribution, and transmission and prevention mechanisms of both COVID-19 and TB.

**Clinical features**: COVID-19 is a highly contagious acute viral disease, whereas TB is a chronic bacterial disease. Both COVID-19 and TB affect the respiratory system, primarily the lungs, and have similar symptoms such as cough, fever and breathing difficulty [17], although the severity and duration of the symptoms are varied. Up to 78% of patients with COVID-19 may be asymptomatic and recover spontaneously [18,19].

**Epidemiology**: TB has long been the leading cause of death due to an infectious disease globally, killing more than 1.5 million people, and with an estimated 10 million new cases in 2018 [20]. The COVID-19 pandemic has now become a public health crisis, and COVID-19 has overtaken TB as the infectious disease killing the most people per day [21]. According to the WHO, as of 8 July 2020, there were more than 11.7 million confirmed cases of COVID-19 and over 540,000 deaths globally. COVID-19 has affected at least 216 countries, areas and territories around the world. High-burden countries for TB have generally experienced a lower incidence of COVID-19 than countries in Europe and North America, although Russia, Brazil, China and India are amongst the top 20 countries for total numbers of cases and deaths due to COVID-19.

**High-risk groups:** Some population groups are at higher risk of developing severe COVID-19 complications [22]. In particular, a higher number of deaths have occurred in adults aged over 60 years [23], in particular men. Similarly, gender differences in TB burden have been reported worldwide, with men more likely to be affected by TB than women [24,25]. Patients with underlying chronic diseases such as hypertension, diabetes, lung cancer and chronic obstructive pulmonary disease are at a higher risk of COVID-19-related death and hospital admission, as well as poor outcomes for TB [17,22,26,27].

**Transmission:** While the exact route of transmission for COVID-19 and TB differ, the major mode of transmission for both diseases is through close contact with infected people [28,29]. For COVID-19, the source of infection can be both symptomatic and asymptomatic patients, while for TB the main source of infection is symptomatic patients with productive cough. The incubation period for TB (from infection to active TB) ranges from several months to two years [30], whereas the incubation period for COVID-19 is approximately 5 days [31].

**Prevention**: Various preventive measures have been taken at global, regional and national levels to reduce the risk of transmission of COVID-19 [32]. The common measures taken by countries to prevent the transmission of the disease include early case detection; prompt isolation of confirmed patients; contact tracing and quarantine of all contacts during the incubation period; social distancing; and communitywide containment, including closure of schools and public facilities, maintaining good hand hygiene through regular washing and use of sanitizers, and wearing of personal protective equipment [33,34]. Many countries have also undertaken strict measures, such as banning of public gatherings, complete lock-down of social and economic activities, and closure of borders to prevent importation of cases [35]. While some countries have been able to control transmission of the disease by implementing the aforementioned interventions, the number of new cases reported continues to rise in many countries at the current time [36]. While there are some vaccine trials under development, there is no evidence that any existing vaccine, including the Bacille Calmette-Guérin vaccine (BCG), protects people against infection with COVID-19 virus.

## 3. The Potential Impact of the COVID-19 Pandemic on Tuberculosis Control

COVID-19 could impact TB control in several ways, including increasing transmission of TB in the household, delaying the diagnosis and treatment of TB and increasing poor treatment outcomes and risks of developing drug-resistant TB. The direct and indirect effects of COVID-19 on national and global economies will have both short-term and long-term consequences for TB programs.

**Impact of COVID-19 on household transmission of TB:** One of the measures undertaken by countries to prevent the spread of COVID-19 is advising or requiring people to stay at home until the situation comes under control [33,34]. While this measure has several advantages in reducing the communitywide transmission of COVID-19, it may also facilitate household transmission of TB. Prolonged contact at household level is one of the risk factors that increases the transmission of TB [37]. A recent modelling study showed that a 3-month lockdown due to COVID-19 would cause an additional 1.65 million TB cases and 438,000 TB deaths in India over the next 5 years [38]. Another study conducted in Brazil showed that intensity of household exposure increased the risk of TB infection and disease among household members [37]. Previous studies have also shown that the prevalence of TB among children in household contact with adult patients is higher than in the general population [39,40], and the risk of household infection is significantly increased with prolonged household contact with sputum positive adults [39,41]. Since TB has a long incubation period, the impact of increased household transmission of TB is only likely to be observed in future years, when an increase in numbers of TB cases may be observed [42,43]. For instance, following the global pandemic of HIV, epidemics of TB were observed in several countries such as South Africa [44,45], which suggests that future public health vigilance is advisable.

**Impact of COVID-19 on TB treatment and diagnostic services:** Overwhelming of health care systems by COVID-19 cases is likely to impact on TB treatment and diagnostic services in several ways: (1) diversion of resources (including human and financial) away from routine services, to manage the pandemic; (2) health service and political leadership, the media and the public focusing on pandemic management and response with limited oversight and accountability of TB programmes; (3) health care personnel experiencing stress and anxiety, key predictors of errors and poor quality of care; (4) health care personnel being required to quarantine, or becoming ill or dying, and therefore not being available for routine services; and (5) stigma and fear of COVID-19 infection at health care facilities, discouraging people from visiting TB services. All of these factors will contribute to delays in the diagnosis and commencement of treatment. As untreated pulmonary TB is the main source of TB infection, late diagnosis and treatment of TB may increase the risk of transmission, especially the household transmission of TB as many people are currently at home. Late diagnosis and inappropriate treatment of TB can also increase the risk of poor treatment outcomes and development of drug-resistant TB. Misdiagnosis and under-detection of TB are ongoing problems for TB programs [4]. It was estimated that globally 3 million TB cases were un-detected in 2018 [4]. This number is likely to increase due the current COVID-19 pandemic.

**Impact of COVID-19 on the prevention and control of TB:** Prevention and control strategies for TB have already been compromised due to the COVID-19 pandemic. Many fora for exchanging TB research and information, such as seminars, workshops and annual conferences, have not been conducted in 2020. For instance, the World Tuberculosis Day, which is celebrated on March 24 each year, to build public awareness about the prevention and control of TB and to raise funding to support TB control efforts, has been cancelled in several countries. Vaccination programs, including the BCG vaccination that has been given to prevent childhood TB, have been negatively affected by COVID-19 [46]. Furthermore, TB preventive therapy, which is often given to high-risk groups to prevent the progression of latent TB to active TB, may also be affected by COVID-19 [47].

The worldwide pandemic of COVID-19 may affect the global strategy of ending TB by 2035 in several ways. Many of the factors affecting diagnostic and testing services also affect prevention and control programmes. Shortages of resources, either directly due to diversion towards pandemic management or indirectly due to broader economic consequences of the pandemic and stretched national budgets, are likely to impact on routine public health programmes. Currently, the attention of the public, government, media and health professions is diverted to COVID-19. Prioritization of TB and other endemic diseases is likely to be less than pre-pandemic levels as a result.

**Impact of COVID-19 on late reactivation of TB:** The impact of COVID-19 on the health status of individuals, including on the functioning of the immune system [48,49], might be associated with a higher risk of developing active TB disease. Pneumonia and respiratory failure caused by COVID-19 might cause long-lasting damage to the respiratory system, particularly the lungs, which might increase the risk of TB [50]. Previous studies have shown that infections with viruses such as HIV and influenza play a role in the development of active TB disease, either directly after exposure to TB or through reactivation of latent TB infection [51,52,53].

Moreover, the COVID pandemic will severely damage global and national economies. The crisis will have a disproportionate impact on the poor, through job losses, loss of remittances, rising prices, and disruptions to services such as education and health care [54]. The World Bank estimates that the global extreme poverty rate could rise by 0.3 to 0.7 percentage points, to around 9 percent in 2020, and around 40 million to 60 million people will fall into extreme poverty in 2020 as a result of COVID-19. This will have a long-term impact on the burden of TB [55], because poverty is widely recognised as an important risk factor for being infected and developing active TB [55,56,57].

## 4. Possible Strategies to Mitigate the Impact of COVID-19 on TB Control

Several strategies can be implemented to mitigate the impact of COVID-19 on TB control (Table 1). For example, to limit household transmission of TB, basic infection prevention and control measures, recommended by the WHO for health care facilities and high-risk settings, can be implemented at home [58]. To avoid TB diagnosis and treatment delay due to COVID-19, use of virtual care and digital health technologies, decentralising TB treatment to community health workers, and supporting private health sectors and academic research institutions to provide TB testing and treatment might all be required.

## 5. Conclusions

The health and economic crisis created by the current COVID-19 pandemic, as well as the public health measures taken to stop the spread of the virus, could have a potential impact on TB prevention and control in many different ways. The proportion of the cumulative disease burden associated with the COVID-19 pandemic due to failures in endemic disease management might end up being greater than that directly caused by COVID-19 itself. It is essential that health systems attempt to maintain routine services for endemic infectious diseases to the highest level possible, recongizing that this may, through necessity, be lower than pre-pandemic levels. It is also essential that health systems have a plan for returning to full service levels as soon as possible, in particular for controlling major endemic diseases such as TB. Economic analyses of the impact of the pandemic should include indirect effects like disruption to routine services and subsequent burden of TB and other endemic infectious diseases. Public health vigilance is necessary to mitigate the impact of COVID-19 on TB prevention and control, with plans in place to manage any increases of TB burden in future years.

## Figures and Tables

**Table 1 tropicalmed-05-00123-t001:** Possible strategies to mitigate the impact of COVID-19 on tuberculosis (TB) control.

Impact of COVID-19 on TB	Strategies to Mitigate the Impact of COVID-19 on TB Control
Increased household transmission of TB	Apply infection prevention and control measures (e.g., cough etiquette, personal protective equipment); Consider using upper-room germicidal ultraviolet (GUV) where indicated; Apply room ventilation (including natural, mixed-mode, mechanical ventilation, and recirculated air through high-efficiency particulate air (HEPA) filters); Separate or isolate people with presumed or demonstrated infectious TB; Provide TB preventive treatment for high-risk groups; Initiate TB treatment early.
Delayed TB diagnosis and treatment services	Maintain supports to essential TB services during and after the COVID-19 pandemic; Provide information to patients about COVID-19 and TB so they can protect themselves and continue their TB treatment; Apply patient-centred delivery of TB prevention, diagnosis, treatment, and care services; Decentralise TB treatment to community health workers and increase access to TB treatment for home-based TB care; Provide adequate supply of TB medication to patients for safe storage at home; Design mechanisms to deliver medicines and to collect specimens for follow-up testing at home; Integrate TB and COVID-19 services for infection control, contact tracing, community-based care, surveillance and monitoring; Provide short-term training for students and health professionals and recruit additional staff to work on TB programs; Change policy if required and support private hospitals, and academic or research centres, to provide TB testing and treatment; Use virtual care and digital health technologies (e.g., video observed therapy) for adherence support, early initiation of treatment, remote monitoring of TB patients, counselling, and follow-up consultations.
Affecting TB prevention and control strategies	Organize virtual conferences, seminars, workshops and fundraising; Design strategies to deliver BCG and TB preventive therapy at home; Create community awareness of the importance of TB services.
Reactivation of TB	Plan additional support and resources to reduce the burden of TB; Conduct research to identify the impact of COVID-19 on reactivation of TB and to design interventions mitigating this problem.

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
