# Peer review of "Impact of the COVID-19 Pandemic on Tuberculosis Control: An Overview"

_tropicalmed, 2020, doi:10.3390/tropicalmed5030123_

Round 1
Reviewer 1 Report
The manuscript submitted by Alene et al. entitled as, “Impacts of the COVID-19 pandemic on tuberculosis control: an overview” provides perspective and impact of the ongoing coronavirus disease pandemic on the management of other diseases of public health concern, with an emphasis on tuberculosis (TB). Authors have nicely provided a detailed understanding of corona virus and its impact on the TB disease management. But, there are few gaps, as summarized below, which should be addressed.
While it is conceivable that such a pandemic had disrupted overall health care management system worldwide, but authors could not come up with a plan on what could be the best policies to manage this situation.
There are several other diseases that might have been impacted in their management. Authors did not provide a rationale why only TB is of importance here. What are authors’ views on other communicable diseases?
The conclusion section is just a summary of what should be done, but it is time for the scientific community to prepare a roadmap that could be followed by the government if such a pandemic is encountered. Authors have failed to provide such details.
Overall, the manuscript is redundant and viewpoint is quite obvious that disturbance in public health care management at this level, which had impacted all economic and public life activities, will entail severe disruptions in various ongoing programs. Without a proper roadmap/plan/outline such a viewpoint is not helpful.
Author Response
Point 1: The manuscript submitted by Alene et al. entitled as, “Impacts of the COVID-19 pandemic on tuberculosis control: an overview” provides perspective and impact of the ongoing coronavirus disease pandemic on the management of other diseases of public health concern, with an emphasis on tuberculosis (TB). Authors have nicely provided a detailed understanding of corona virus and its impact on the TB disease management. But, there are few gaps, as summarized below, which should be addressed.
Response 1: We are grateful for the thoughtful comments of the reviewers. We have carefully reviewed the comments and have revised the manuscript accordingly.
Point 2: While it is conceivable that such a pandemic had disrupted overall health care management system worldwide, but authors could not come up with a plan on what could be the best policies to manage this situation.
Response 2: we have highlighted relevant policy recommendations in the conclusion sections of the manuscript. In addition, we have now included possible strategies that could help to mitigate the impact of COVID-19 on TB control, in a summary table.
Possible strategies to mitigate the impact of COVID-19 on TB |
|
Impact of COVID-19 on TB |
Strategies to mitigate the impact of COVID-19 on TB programs |
Increased household transmission of TB
|
· Apply infection prevention and control measures (e.g. cough etiquette, personal protective equipment) · Consider using upper-room germicidal ultraviolet (GUV) where indicated · Apply room ventilation (including natural, mixed-mode, mechanical ventilation, and recirculated air through high-efficiency particulate air [HEPA] filters) · Separate or isolate people with presumed or demonstrated infectious TB · Provide TB preventive treatment for high risk groups · Early initiation of effective TB treatment |
Delayed TB diagnosis and treatment services |
· Provide short-term training for students and health professionals and recruit additional staff to work on TB programs · Change policy if required and support private hospital, academic or research centres to provide TB testing and treatment · Integrate TB and COVID-19 services for infection control, contact tracing, community-based care, surveillance, and monitoring · Decentralise TB treatment to community health workers and increase access to TB treatment for home-based TB care. · Apply patient-centred delivery of TB prevention, diagnosis, treatment, and care services · Provide adequate supply of TB medication to patients for safe storage at home · Design mechanisms to deliver medicines and to collect specimens for follow-up testing at home · Health authorities should maintain support to essential TB services · Provide information to patients about COVID-19 and TB to protect themselves and continue with their TB treatment · Using virtual care and digital health technologies (e.g. video observed therapy) for adherence support, early initiation of treatment, remote monitoring of TB patients, counselling, and follow-up consultations |
Affecting TB prevention and control strategies
|
· Organize virtual conferences, seminars, workshops, and fundraising · Design strategies to deliver BCG and TB preventive therapy at home · Create community awareness of the importance of TB services |
Late reactivation of TB |
· Conduct research to identify the impact of COVID-19 on reactivation of TB and to design interventions mitigating this problem · Plan additional support and resources to reduce the burden of TB |
Point 3: There are several other diseases that might have been impacted in their management. Authors did not provide a rationale why only TB is of importance here. What are authors’ views on other communicable diseases?
Response 3: The impacts of COVID-19 on other diseases such as cancer and diabetes have been addressed in recent reviews [14, 15]. However, there is no review on the impact of COVID-19 on TB. TB is the leading cause of death due to an infectious disease globally and it is anticipated that people ill with both TB and COVID-19 may have poorer treatment outcomes. Most importantly, the public health response to COVID-19 to isolate people in their homes for extended periods will facilitate transmission of TB as close household contact, particularly in low-socioeconomic and overcrowded conditions, is a key risk factor for TB. Additionally, both diseases attack primarily the lungs and show similar symptoms such as cough, fever and difficulty breathing, although the duration and severity of the symptoms are varied. In addition, both biological agents transmit mainly via close contact. Therefore, understanding the potential impacts of COVID-19 on TB would be important for designing prevention strategies.
Point 4: The conclusion section is just a summary of what should be done, but it is time for the scientific community to prepare a roadmap that could be followed by the government if such a pandemic is encountered. Authors have failed to provide such details.
Response 4: We believe that our overview is important to stimulate development of roadmaps to support both researchers and governments. In response 2 we have now outlined what such a roadmap might contain.
Point 5: Overall, the manuscript is redundant, and viewpoint is quite obvious that disturbance in public health care management at this level, which had impacted all economic and public life activities, will entail severe disruptions in various ongoing programs. Without a proper roadmap/plan/outline such a viewpoint is not helpful.
Response 5: We believe that understanding the potential impacts of COVID-19 on TB programs would be helpful to design further research and intervention programs.
Reviewer 2 Report
The authors provide a nice explanation of impact of secondary diseases on TB. This is critical for low income section or countries were TB is still relevant. Good read. The manuscript may be accepted.
Author Response
Point 1: The authors provide a nice explanation of impact of secondary diseases on TB. This is critical for low income section or countries were TB is still relevant. Good read. The manuscript may be accepted.
Response 5: We are grateful for the thoughtful comments of the reviewers.
Reviewer 3 Report
It is reasonable to think that the worldwide COVID19 situation will have great impact on the re-insurgence of several diseases and health problems, and therefore probably also for TB, among others. However, it is not clear why COVID19 should have an impact particularly on TB. This disease is influenced by several social and environmental factors. Clearly the level of poverty on which people lives are important for TB, both for the hygienic conditions of life (promiscuity, access to clean water, etc.) and for the access to adequate health services. Therefore, the low-income countries are those more experiencing BT problems as well as those strata of population affected by poverty and social-economic difficulties in the other countries. In addition, wars, human migrations and other global factors have been proved to influence the occurrence of TB. Therefore, the effect of COVID19 should be analysed in context of all the other existing drivers. The authors report some descriptive reasoning, quite plausible, but not supported by data analysis. For example, an analysis of the countries where the various factors, including COVID19, might have a synergic impact on TB could be more informative than the simple speculations reported in this paper.
Author Response
Point 1: It is reasonable to think that the worldwide COVID19 situation will have great impact on the re-insurgence of several diseases and health problems, and therefore probably also for TB, among others. However, it is not clear why COVID19 should have an impact particularly on TB.
Response 1: Refer to Response 3, reviewer 1:
The impacts of COVID-19 on other diseases such as cancer and diabetes have been addressed in recent reviews [14, 15]. However, there is no review on the impact of COVID-19 on TB. TB is the leading cause of death due to an infectious disease globally and it is anticipated that people ill with both TB and COVID-19 may have poorer treatment outcomes. Most importantly, the public health response to COVID-19 to isolate people in their homes for extended periods will facilitate transmission of TB as close household contact, particularly in low-socioeconomic and overcrowded conditions, is a key risk factor for TB. Additionally, both diseases attack primarily the lungs and show similar symptoms such as cough, fever and difficulty breathing, although the duration and severity of the symptoms are varied. In addition, both biological agents transmit mainly via close contact. Therefore, understanding the potential impacts of COVID-19 on TB would be important for designing prevention strategies.
Point 2: This disease is influenced by several social and environmental factors. Clearly the level of poverty on which people lives are important for TB, both for the hygienic conditions of life (promiscuity, access to clean water, etc.) and for the access to adequate health services. Therefore, the low-income countries are those more experiencing BT problems as well as those strata of population affected by poverty and social-economic difficulties in the other countries. In addition, wars, human migrations and other global factors have been proved to influence the occurrence of TB. Therefore, the effect of COVID19 should be analysed in context of all the other existing drivers.
Response 2: We acknowledged that the impacts of COVID-19 may vary depending on several individual-level and ecological-level factors. The influence of such factors can be quantified using primary or secondary data. However, the current study is an overview without primary or secondary data, and it would have some limitations as it is impossible to be exhaustive in this short overview.
Point 3: The authors report some descriptive reasoning, quite plausible, but not supported by data analysis. For example, an analysis of the countries where the various factors, including COVID19, might have a synergic impact on TB could be more informative than the simple speculations reported in this paper.
Response 3: We have thoroughly revised the manuscript to address this comment.
Reviewer 4 Report
General comments:
Alene and colleagues present an interesting and balanced review of the likely impact of the current pandemic on TB control. The authors cover most of the major issues in this important area, although it is impossible to be exhaustive. I generally thought that there might be a few more ideas that could be incorporated if the current text is made a little more succinct. For example, are there any potential synergies? – is this a time for advocacy to improve health systems in developing countries and strengthen medical research?, etc. However, I do not wish to be prescriptive on these points, which I will leave to the authors.
The writing is generally fairly clear and readable, but there are quite a number of places where it could be improved – and I would encourage the authors to work through the text a couple more times. An article of this type would be considerably stronger if the sentence flow is really absolutely fluent and enjoyable to read.
Comments by section:
Introduction
This section focuses largely on the history of disruptions to TB services. I think this is a really nice lead in to the area, but I would just suggest changing the heading of this section to “Historical perspective” or similar (because the section isn’t really a general introduction). (If allowed by journal rules.)
Paragraph beginning line 34. I’m not convinced that we know that TB was well controlled globally in the post-war era. Until 1990, we had extremely limited information on TB burden in LMICs. Consider revising to indicate that TB was well controlled in high-income settings.
The coronavirus disease 2019 (COVID-19) pandemic
I feel this section would be much improved by stressing the relevance to TB. Currently, it is mostly (although not entirely) a generic description of COVID-19. This is interesting, but there are a huge number of summaries of the literature in this area already available, so I think there needs to be a greater focus on what the particularly relevant messages are here – consider issues such as: how is Covid like TB?, why are they different?, how might the features of one help understand or fight the other?
The potential impacts of the COVID-19 pandemic on tuberculosis control
This seems a higher tier of sub-heading than the following four sub-headings.
Impact of COVID-19 on household transmission of TB
Perhaps worth explaining why any positive effects of lockdown are unlikely to significantly decrease TB burden. Consider mentioning contact saturation – which might mitigate these effects (e.g. McCreesh et al., Ragonnet et al.).
Impacts of COVID-19 on the prevention and control of TB
Consider mentioning the specific issues around BCG vaccination being used for its unproven effects in preventing Covid.
Impacts of COVID-19 on the long-term development of TB
Unclear what the term “long-term development” refers to. Perhaps the authors are meaning “late reactivation”.
Minor comments:
Line 18 – remove “the” preceding “low- and middle-income”
Line 22 – suggest prefer first person plural to passive voice
Line 38 – replace “implementations” with “implementation”
Sentence beginning line 47 – suggest move “and no catastrophic costs due to TB by 2035” to end of sentence, because this target does not need to be compared to 2015.
Line 52 – suggest add comma after “pandemic”
Sentence beginning line 54 – I generally prefer to just indicate “we reviewed” (because the authors did review this area), rather than saying that reviewing the area was the aim
Some inconsistency with date formatting.
Line 82 – replace “underline” with “underlying”
Line 90 – consider whether to replace “not” with “unlikely to be”
Line 92 – replace “preventative” with “preventive”
Line 94 – probably OK, but for Covid in particular, sometimes viral shedding occurs before symptoms (so possibly before the person becomes a “case”) – perhaps there is a way to word this sentence to avoid that possible confusion
Line 97 – place comma before “as well as”
Line 99 – place comma before “such as”
Line 101 – remove “the” before “transmission”
Line 105 – remove “the” before “transmission”
Line 108 – when discussing future impacts, suggest “is likely to” rather than “will”
Line 137 – hyphenate compound adjective “drug resistant”
Line 137 – replace “Miss-diagnosis” with “mis-diagnosis”
Line 142 – sentence beginning “Many…” needs commas
Line 173 – add commas around “as well as … the virus”
Line 183 – remove “potential” – only actual risks need to be mitigated
No clear reason why impact(s) is sometimes singular and sometimes plural
Author Response
General comments: Alene and colleagues present an interesting and balanced review of the likely impact of the current pandemic on TB control. The authors cover most of the major issues in this important area, although it is impossible to be exhaustive.
Point 1: I generally thought that there might be a few more ideas that could be incorporated if the current text is made a little more succinct. For example, are there any potential synergies? – is this a time for advocacy to improve health systems in developing countries and strengthen medical research?, etc. However, I do not wish to be prescriptive on these points, which I will leave to the authors.
Response 1: Although TB is more common in developing countries, COVID-19 has affected both developed and developing countries.
Point 2: The writing is generally fairly clear and readable, but there are quite a number of places where it could be improved – and I would encourage the authors to work through the text a couple more times. An article of this type would be considerably stronger if the sentence flow is really absolutely fluent and enjoyable to read.
Response 2: We have revised the manuscript thoroughly to address this comment.
Comments by section:
Point 3: Introduction
This section focuses largely on the history of disruptions to TB services. I think this is a really nice lead in to the area, but I would just suggest changing the heading of this section to “Historical perspective” or similar (because the section isn’t really a general introduction). (If allowed by journal rules.)
Response 3: As suggested by the reviewer, the heading is now changed to “Historical perspective”.
Point 4: Paragraph beginning line 34. I’m not convinced that we know that TB was well controlled globally in the post-war era. Until 1990, we had extremely limited information on TB burden in LMICs. Consider revising to indicate that TB was well controlled in high-income settings.
Response 4: this is also corrected
Point 5: The coronavirus disease 2019 (COVID-19) pandemic
I feel this section would be much improved by stressing the relevance to TB. Currently, it is mostly (although not entirely) a generic description of COVID-19. This is interesting, but there are a huge number of summaries of the literature in this area already available, so I think there needs to be a greater focus on what the particularly relevant messages are here – consider issues such as: how is Covid like TB?, why are they different?, how might the features of one help understand or fight the other?
Response 5: This section is thoroughly revised to address the reviewer’s comments.
Point 6: The potential impacts of the COVID-19 pandemic on tuberculosis control
This seems a higher tier of sub-heading than the following four sub-headings.
Response 6: Yes, this is a higher tier of the sub-headings, which is now adjusted to reflect this.
Point 7: Impact of COVID-19 on household transmission of TB
Perhaps worth explaining why any positive effects of lockdown are unlikely to significantly decrease TB burden. Consider mentioning contact saturation – which might mitigate these effects (e.g. McCreesh et al., Ragonnet et al.).
Response 7: “Although prolonged contact at household level may result in contact saturation, it is unlikely to decrease TB burden [1, 2]. A recent modelling study showed that a 3-month lockdown due to COVID-19 would cause an additional 1.65 million TB cases and 438,000 TB deaths in India over the next 5 years [3]”. This information now included in the revised version of the manuscript.
Point 8: Impacts of COVID-19 on the prevention and control of TB
Consider mentioning the specific issues around BCG vaccination being used for its unproven effects in preventing COVID.
Response 8: “While there are some vaccine trials under development, there is no evidence that any vaccine including the Bacille Calmette-Guérin vaccine (BCG) protects people against infection with COVID-19 virus.” This information is also included in the revised version of the manuscript.
Point 9: Impacts of COVID-19 on the long-term development of TB
Unclear what the term “long-term development” refers to. Perhaps the authors are meaning “late reactivation”.
Response 9: This now corrected
Minor comments:
Point 10: Line 18 – remove “the” preceding “low- and middle-income”
Response 10: it is now corrected
Point 11: Line 22 – suggest prefer first person plural to passive voice
Response 11: We have corrected the sentence as suggested
Point 12: Line 38 – replace “implementations” with “implementation”
Response 12:it is now corrected
Point 13: Sentence beginning line 47 – suggest move “and no catastrophic costs due to TB by 2035” to end of sentence, because this target does not need to be compared to 2015.
Response 13: this is also corrected
Point 14: Line 52 – suggest add comma after “pandemic”
Response 14: this is also corrected.
Point 15: Sentence beginning line 54 – I generally prefer to just indicate “we reviewed” (because the authors did review this area), rather than saying that reviewing the area was the aim. Some inconsistency with date formatting.
Response 15: We have now corrected both the sentence and date formatting.
Point 16: Line 82 – replace “underline” with “underlying”
Response 16: it is now corrected
Point 17: Line 90 – consider whether to replace “not” with “unlikely to be”
Response 17: this also corrected
Point 18: Line 92 – replace “preventative” with “preventive”
Response 18: it is now corrected
Point 19: Line 94 – probably OK, but for COVID in particular sometimes viral shedding occurs before symptoms (so possibly before the person becomes a “case”) – perhaps there is a way to word this sentence to avoid that possible confusion
Response 19: it also now corrected
Point 20: Line 97 – place comma before “as well as”
Response 20: it is now corrected
Point 21: Line 99 – place comma before “such as”
Response 21: comma is now added.
Point 22: Line 101 – remove “the” before “transmission”
Response 22: it is now removed
Point 23: Line 105 – remove “the” before “transmission”
Response 23:
Point 24: Line 108 – when discussing future impacts, suggest “is likely to” rather than “will”
Response 24: it is now corrected
Point 25: Line 137 – hyphenate compound adjective “drug resistant”
Response 25: it is also corrected
Point 26: Line 137 – replace “Miss-diagnosis” with “mis-diagnosis”
Response 26: this is also corrected
Point 27: Line 142 – sentence beginning “Many…” needs commas
Response 27: comma is now added
Point 28: Line 173 – add commas around “as well as … the virus”
Response 28: comma is now added
Point 29: Line 183 – remove “potential” – only actual risks need to be mitigated
Response 29: the word “potential” is now removed
Point 30: No clear reason why impact(s) is sometimes singular and sometimes plural
Response 30: this is corrected throughout the manuscript
References
- Ragonnet R, Trauer JM, Geard N, Scott N, McBryde ES. Profiling Mycobacterium tuberculosis transmission and the resulting disease burden in the five highest tuberculosis burden countries. BMC medicine. 2019;17(1):208.
- McCreesh N, White RG. An explanation for the low proportion of tuberculosis that results from transmission between household and known social contacts. Scientific reports. 2018;8(1):5382.
- Cilloni L, Fu H, Vesga JF, Dowdy D, Pretorius C, Ahmedov S, et al. The potential impact of the COVID-19 pandemic on tuberculosis: a modelling analysis. medRxiv. 2020.
Round 2
Reviewer 1 Report
Revised manuscript is in a much better shape. Roadmap plan in the form of a table is outstanding. Authors efforts are appreciated.
Few minor mistakes:
There are few typos, specially in the newly added sections (e.g. Line 24).
Similarly, font sizes are all over the place.
For comparision of COVID-19 and TB (line 70..), few lines on the scope of paragraph should be added. It is not very apparent that authors are going to compare various aspects of two different diseases. Similarly, this section should be summarized briefly, comparing similarity and why TB should also be given importance while dealing with COVID-19.
Author Response
Point 1: Revised manuscript is in a much better shape. Roadmap plan in the form of a table is outstanding. Authors efforts are appreciated.
Response 1: We are grateful for the thoughtful comments of the reviewers.
Point 2: Few minor mistakes: There are few typos, especially in the newly added sections (e.g. Line 24).
Response 2: This is now corrected
Point 3: Similarly, font sizes are all over the place.
Response 3: The font sizes are now adjusted.
Point 4: For comparison of COVID-19 and TB (line 70..), few lines on the scope of paragraph should be added. It is not very apparent that authors are going to compare various aspects of two different diseases. Similarly, this section should be summarized briefly, comparing similarity and why TB should also be given importance while dealing with COVID-19.
Response 4: We have now included a summary to show the scope of the section.
Reviewer 3 Report
Thank the authors for the changes and amendments to the text which is more clear and complete
Author Response
Point 1: Thank the authors for the changes and amendments to the text which is more clear and complete
Response 1: Thank you for reviewing our manuscript for the second time.
Reviewer 4 Report
General comments
Many thanks to the authors for responding to my comments so carefully.
There are no major outstanding issues with this paper, although the language and flow of the text could still be improved. I have listed a number of points at which improvements could be made, but my making specific comments about individual sentences on the PDF may not be the most efficient way of improving the writing. Therefore, I will leave it to the editors as to how this can be best and most efficiently achieved.
The authors may also be aware that the following paper, which may be relevant, was published in the last few days: Hogan et al. “Potential impact of the COVID-19 pandemic on HIV, tuberculosis, and malaria in low-income and middle-income countries: a modelling study” Lancet Global Health. (I actually don’t think there is any necessity to cite this study, which makes a lot of unproven assumptions – but just draw it to the authors’ attention in case they wish to.)
Minor points:
Throughout manuscript – inconsistent font and formatting, hyphenate “high-risk”
Table 1 – very inconsistent in the parts of speech used, which makes the Table difficult to read
Table 1 – in “Delayed TB diagnosis and treatment services”, seems strange to list the second bullet before the eighth. Presumably the focus should be on maintaining essential services first, with other services enlisted to support if this cannot be achieved.
Line 21 – no need to include both “may have” and “potential”
Line 24 – space missing in “aspossible”
Line 52 – space missing in “anda”
Line 63 – suggest change “will” to “could” (because contact saturation could mitigate this effect) and/or revise wording “will facilitate transmission of TB as close household contact” – not clear what this means. That is, while I agree that isolating at home will facilitate household transmission, I’m not certain it will increase transmission overall – and I’m not sure which statement the authors are making here.
Line 68 – “Therefore, we reviewed to present the potential impact” – suggest revise, not sure exactly what is meant here. Perhaps “to present” not needed.
Line 85 – “groups of the population” can be shortened to “population groups”
Line 88 – remove “are”
Line 92 – “are not the same” can be shortened to “differ”
Line 95 – missing full stop
Lines 109-110 – missing commas around “the Bacille Calmette-Guérin vaccine (BCG)”
Lines 131-132 – revise wording and/or run these two short sentences together
Line 136 – change “focussing” to “focusing”
Lines 177-178 – revise wording and/or run these two short sentences together
Line 183 – revise “delaying in”
Line 194 – no need for both “may have” and “potential” in one sentence
Lines 195-197 – “the” missing from sentence and sentence sounds too conversational
Author Response
Point 1: Many thanks to the authors for responding to my comments so carefully. There are no major outstanding issues with this paper, although the language and flow of the text could still be improved. I have listed a number of points at which improvements could be made, but my making specific comments about individual sentences on the PDF may not be the most efficient way of improving the writing. Therefore, I will leave it to the editors as to how this can be best and most efficiently achieved.
Response 1: We appreciate the thoughtful review and constructive suggestions of the reviewer. We have carefully reviewed the comments and have revised the manuscript accordingly.
Point 2: The authors may also be aware that the following paper, which may be relevant, was published in the last few days: Hogan et al. “Potential impact of the COVID-19 pandemic on HIV, tuberculosis, and malaria in low-income and middle-income countries: a modelling study” Lancet Global Health. (I don’t think there is any necessity to cite this study, which makes a lot of unproven assumptions – but just draw it to the authors’ attention in case they wish to.)
Response 2: We thank you for the reviewer for bringing this paper to our attention.
Minor points:
Point 3: Throughout manuscript – inconsistent font and formatting, hyphenate “high-risk”
Response 3: The font size and the formatting are now corrected
Point 4: Table 1 – very inconsistent in the parts of speech used, which makes the Table difficult to read
Response 4: The table is now revised to be more consistent.
Point 5: Table 1 – in “Delayed TB diagnosis and treatment services”, seems strange to list the second bullet before the eighth. Presumably the focus should be on maintaining essential services first, with other services enlisted to support if this cannot be achieved.
Response 5: This is now revised.
Point 6: Line 21 – no need to include both “may have” and “potential”
Response 6: This is now corrected
Point 7: Line 24 – space missing in “aspossible”
Response 6: This also corrected
Point 8: Line 52 – space missing in “anda”
Response 8: This is also corrected
Point 9: Line 63 – suggest change “will” to “could” (because contact saturation could mitigate this effect) and/or revise wording “will facilitate transmission of TB as close household contact” – not clear what this means. That is, while I agree that isolating at home will facilitate household transmission, I’m not certain it will increase transmission overall – and I’m not sure which statement the authors are making here.
Response 9: This is now corrected
Point 10: Line 68 – “Therefore, we reviewed to present the potential impact” – suggest revise, not sure exactly what is meant here. Perhaps “to present” not needed.
Response 10: This also now corrected
Point 11: Line 85 – “groups of the population” can be shortened to “population groups”
Response 21: This is now corrected
Point 12: Line 88 – remove “are”
Response 13: It is now removed
Point 13: Line 92 – “are not the same” can be shortened to “differ”
Response 14: This is now corrected
Point 14: Line 95 – missing full stop
Response 15: It is now corrected
Point 15: Lines 109-110 – missing commas around “the Bacille Calmette-Guérin vaccine (BCG)”
Response 16: This is now corrected
Point 16: Lines 131-132 – revise wording and/or run these two short sentences together
Response 17: This sentence is now revised
Point 17: Line 136 – change “focussing” to “focusing”
Response 18: This is also corrected
Point 18: Lines 177-178 – revise wording and/or run these two short sentences together
Response 19: It is now corrected
Point 19: Line 183 – revise “delaying in”
Response 20: It is now correct
Point 20: Line 194 – no need for both “may have” and “potential” in one sentence
Response 1: This is now corrected
Point 21: Lines 195-197 – “the” missing from sentence and sentence sounds too conversational
Response 21: This is now corrected